# Scalable and Differentiable Point-Cloud Registration Using Maximum Mean Discrepancy

**Rixon Crane** [1]   **Fahira Afzal Maken** [2]   **Nicholas Lawrance** [1]   **Stanislav Funiak** [3]   **Kasra Khosoussi** [4]   **Ming Xu** [5]
**Russell Tsuchida** [6]

## Abstract

We present MMD-Reg, a novel correspondence-free approach to point-cloud registration that is differentiable and has linear computational complexity in the number of points. We model registration as a nonlinear least-squares problem based on the Maximum Mean Discrepancy, approximated using random Fourier features. The resulting objective can be solved efficiently with standard methods such as Levenberg–Marquardt, and the solution is differentiable via the implicit function theorem. This allows MMD-Reg to be used as a differentiable optimization layer within end-to-end trainable models, supporting registration under challenging conditions such as poor initial alignment and partial overlap. We demonstrate this Neural MMD-Reg formulation by integrating the layer with a set transformer, training the resulting model in supervised and unsupervised settings, and comparing its performance against recent learning-based methods. We also evaluate standalone MMD-Reg, comparing its accuracy and scalability against widely used non-learning-based registration methods.

## 1. Introduction

Rigid 3D point-cloud registration is a fundamental problem in computer vision and robotics, where the goal is to find a rigid transformation (rotation and translation) that aligns two sets of 3D points (point clouds). Accurate and efficient registration is used in 3D object reconstruction, naviga-

tion for robots and autonomous vehicles (Pomerleau et al., 2015; Brightman et al., 2023), construction site monitoring (Kim et al., 2018), and provides pose odometry estimates for LiDAR- and depth-based simultaneous localization and mapping (SLAM) systems (Carlone et al., 2025; Ebadi et al., 2023). In real-world settings, registration algorithms should scale to large environments and be robust to sensor noise, outliers, occlusions, and non-uniform sampling densities that may arise (Yuan et al., 2025). Furthermore, modern pipelines may favor differentiable solutions that can be embedded in end-to-end trainable systems (Pineda et al., 2022). Many existing methods trade off among scalability, robustness, and differentiability. Moreover, they may depend on preprocessing for accuracy and computational tractability, such as downsampling and outlier filtering (Zhu et al., 2023; Dong et al., 2025). While effective in practice, these preprocessing stages introduce additional complexity and hyperparameter sensitivity.

One of the most widely used registration algorithms is Iterative Closest Point (ICP) (Besl & McKay, 1992; Chen & Medioni, 1992), which frames registration as a geometric least-squares problem. At each iteration, ICP establishes correspondences via nearest-neighbor search and estimates the rigid transformation that minimizes either point-to-point (Besl & McKay, 1992) or point-to-plane (Chen & Medioni, 1992) residuals. Nearest-neighbor queries can be accelerated with $k$-d trees (Rusinkiewicz & Levoy, 2001), allowing ICP to scale to large point clouds. Much work has refined ICP to improve speed and robustness, such as applying robust loss functions (Babin et al., 2019) and using covariance matrices to model local geometric structure (Segal et al., 2009). Together with high-performance implementations, these variants make ICP an effective choice in many real-time systems when paired with appropriate preprocessing.

Careful implementation and preprocessing are often required for geometric registration methods that rely on hard nearest-neighbor correspondences and quadratic point-to-point residuals, such as ICP. Measurement noise, outliers, and oversampled regions induced by non-uniform sampling can exert disproportionate influence on the objective, steering the optimizer toward spurious matches and local

---

[1]Data61, CSIRO, Brisbane, Australia [2]Data61, CSIRO, Sydney, Australia [3]Work done while at Data61, CSIRO, Brisbane, Australia [4]School of Electrical Engineering and Computer Science, The University of Queensland, Brisbane, Australia [5]CVLab, EPFL, Lausanne, Switzerland [6]Department of Data Science & AI, Monash University, Melbourne, Australia. Correspondence to: Rixon Crane <rixon.crane@data61.csiro.au>.

*Proceedings of the 43rd International Conference on Machine Learning*, Seoul, South Korea. PMLR 306, 2026. Copyright 2026 by the author(s).

minima (Yang et al., 2016; Yuan et al., 2025). Moreover, nearest-neighbor assignments are discrete and can change abruptly between iterations, making the overall solution non-differentiable. They can also be inefficient on GPUs due to irregular memory access patterns and branching (Nam et al., 2016; Koide et al., 2021). In contrast, probabilistic registration methods often improve robustness by comparing point clouds through distributional representations and replacing hard correspondences with soft associations, though often at the cost of increased per-iteration complexity (Gao & Tedrake, 2019).

Probabilistic registration methods often model point clouds as samples from latent spatial distributions and estimate alignment by optimizing a statistical objective under a parametric model. A representative example is Coherent Point Drift (CPD) (Myronenko & Song, 2010), which casts registration as density estimation by treating one point cloud as the centroids of a Gaussian mixture model (GMM) and the other as noisy observations. Using Expectation-Maximization (EM) (Dempster et al., 1977), CPD jointly infers soft correspondences and the transformation by maximizing the likelihood. Other popular GMM-based methods include GMMReg (Jian & Vemuri, 2011), which represents both clouds as GMMs and minimizes a closed-form discrepancy between mixtures, and FilterReg (Gao & Tedrake, 2019), a scalable EM formulation related to CPD in which the E-step is implemented via efficient Gaussian filtering on a permutohedral lattice.

In this paper, we present a novel registration formulation, detailed in Section 3, in which rigid alignment is posed as minimization of a random-feature surrogate of the Maximum Mean Discrepancy (MMD) (Gretton et al., 2012). Rather than fitting a parametric model to latent spatial distributions, we estimate the discrepancy directly from samples (the point clouds) using MMD. MMD is a non-parametric discrepancy between probability measures that requires no explicit density models and can be expressed as the distance between their kernel mean embeddings in a reproducing kernel Hilbert space (RKHS). Given two point clouds, the standard (biased) empirical estimator of MMD has quadratic complexity in the number of points. To scale to large datasets, we approximate the kernel with random Fourier features (RFF) (Rahimi & Recht, 2007), reducing computation to linear in the number of points for a fixed number of features. This yields a smooth nonlinear least-squares objective over the rigid transformation, which we solve efficiently using the Levenberg–Marquardt algorithm (Moré, 1978). The resulting objective is non-convex, so in standalone form MMD-Reg should be viewed as a local registration method. In the experimental results (Section 4), harder alignment regimes are handled by sequential refinement over the kernel scale or by learned initialization in Neural MMD-Reg.

Because our formulation is correspondence-free, alignment is driven by distributional discrepancy rather than brittle pairwise matches. Moreover, with characteristic kernels, MMD is a metric on probability measures (Gretton et al., 2012), encouraging faithful alignment of underlying densities even in the presence of noise and outliers. Our approach differs from prior smoothing, filtering, and coarse-to-fine registration methods in that the registration objective is derived directly from a random-feature surrogate of MMD.

An additional benefit of our formulation is that the solution can be differentiated efficiently via the implicit function theorem (IFT) (Krantz & Parks, 2003). Consequently, our approach can be embedded as a differentiable optimization layer, supporting high inference speed on GPUs within end-to-end trainable models. Combining the strengths of differentiable optimization layers and deep learning has proven effective across a wide range of tasks (Pineda et al., 2022). Several libraries support such layers, including Deep Declarative Networks (DDNs) (Gould et al., 2022), Theseus (Pineda et al., 2022), JAXopt (Blondel et al., 2022), and TorchOpt (Ren et al., 2023). Our aim is not to replace global-plus-local pipelines, but to provide a scalable, differentiable, correspondence-free local objective that is useful when registration is embedded in a learned system or when a coarse initialization is available. In Section 4.3, we couple our optimization layer with a set transformer (Lee et al., 2019) and train the model in supervised and unsupervised settings on ModelNet40 (Wu et al., 2015). We complement this with standalone evaluations of MMD-Reg, without a learned initializer, on synthetic and real outdoor data in Sections 4.1 and 4.2. Code is available at `https://github.com/csiro-robotics/mmd-reg`.

## 2. Background

We assume the source point cloud $X = \{x_i\}_{i=1}^m$ and the target point cloud $Y = \{y_j\}_{j=1}^n$ are i.i.d. samples from unknown spatial distributions $P$ and $Q$, respectively. We seek a rigid transformation $T \in SE(3)$ that minimizes an empirical estimate of a discrepancy between $P$ and $Q$. We choose MMD because it is non-parametric, requires no explicit density models, and, with characteristic kernels, is a metric on probability measures.

**Maximum mean discrepancy.** Let $P$ and $Q$ be Borel probability measures on the topological space $\mathcal{X} = \mathbb{R}^3$, representing the distributions from which $X$ and $Y$ are sampled. Let $\mathcal{H}$ be the reproducing kernel Hilbert space (RKHS) associated with a symmetric positive semidefinite (PSD) kernel $k : \mathcal{X} \times \mathcal{X} \to \mathbb{R}$. MMD is defined by

$$\mathrm{MMD}(P, Q) := \sup_{\substack{f \in \mathcal{H} \\ \|f\|_{\mathcal{H}} \leq 1}} \left( \mathbb{E}_{x \sim P}\big[f(x)\big] - \mathbb{E}_{y \sim Q}\big[f(y)\big] \right).$$

(1)

Because $\mathcal{H}$ is an RKHS, we can express MMD in terms of kernel mean embeddings under mild conditions on $k$. Let $\phi$ be the canonical feature map associated with $k$, namely $\phi(x) = k(x, \cdot)$. For all $f$ in $\mathcal{H}$, we have the reproducing property $f(x) = \langle f, \phi(x) \rangle_{\mathcal{H}}$. If $k$ is measurable and bounded in expectation, then the mean embeddings $\mu_P := \mathbb{E}_{x \sim P}[\phi(x)]$ and $\mu_Q := \mathbb{E}_{y \sim Q}[\phi(y)]$ are in $\mathcal{H}$, and

$$\text{MMD}(P, Q) = \|\mu_P - \mu_Q\|_{\mathcal{H}}, \tag{2}$$

by applying the reproducing property and Cauchy–Schwarz inequality to (1) (Gretton et al., 2012). Furthermore, if $k$ is a characteristic kernel, then $\text{MMD}(P, Q) = 0$ if and only if $P = Q$, implying MMD is a metric on the Borel probability measures on $\mathcal{X}$ (Gretton et al., 2012). In practice, characteristic kernels ensure that minimizing MMD aligns the distributions beyond just their first few moments, which is desirable when point clouds are noisy or contain outliers.

**Empirical estimation.** The RKHS structure lets us write $\text{MMD}^2$ in terms of kernel expectations, which we can approximate by sample averages on the observed point clouds $X$ and $Y$. In particular, for the canonical feature map $\phi(x)$ we have $\langle \phi(x), \phi(y) \rangle_{\mathcal{H}} = k(x, y)$, thus (2) gives

$$\text{MMD}^2(P, Q) = \mathbb{E}_{x, x' \sim P}[k(x, x')] + \mathbb{E}_{y, y' \sim Q}[k(y, y')] \\ - 2\,\mathbb{E}_{x \sim P, y \sim Q}[k(x, y)]. \tag{3}$$

Replacing the population expectations in (3) by their sample averages gives the (biased) empirical estimator

$$\widehat{\text{MMD}}^2(X, Y) = \frac{1}{m^2} \sum_{i,j=1}^{m} k(x_i, x_j) + \frac{1}{n^2} \sum_{i,j=1}^{n} k(y_i, y_j) \\ - \frac{2}{mn} \sum_{i=1}^{m} \sum_{j=1}^{n} k(x_i, y_j). \tag{4}$$

**Fast random feature methods.** The empirical estimator (4) scales quadratically in the number of points. To obtain a linear-scaling surrogate, we follow (Rahimi & Recht, 2007) and approximate $k$ using RFF, yielding a finite-dimensional feature map $z : \mathcal{X} \to \mathbb{R}^{2D}$ such that $k(x, y) \approx z(x)^\top z(y)$. When $k$ is continuous, positive-definite, and shift-invariant, this construction follows from Bochner's theorem, which implies that $k$ is the Fourier transform of a finite non-negative measure (and conversely). When normalized so that $k(x, x) = 1$, this measure corresponds to a probability density $p(\omega)$. This also gives the approximation a spectral interpretation: the sampled frequencies define a band-limited representation of the point distributions, and the kernel scale controls the coarse-to-fine spatial structure emphasized by the objective. For our method, we use the

real-valued random-feature map $z : \mathcal{X} \to \mathbb{R}^{2D}$ defined by

$$z(x) = \begin{bmatrix} z_1(x) \\ z_2(x) \\ \vdots \\ z_{2D}(x) \end{bmatrix} := \frac{1}{\sqrt{D}} \begin{bmatrix} \cos(\omega_1^\top x) \\ \sin(\omega_1^\top x) \\ \vdots \\ \cos(\omega_D^\top x) \\ \sin(\omega_D^\top x) \end{bmatrix}, \tag{5}$$

with frequencies $\omega_i$ sampled i.i.d. from $p(\omega)$. This construction has lower variance (for the squared exponential kernel) than the cosine-only variant with random phase (Sutherland & Schneider, 2015). In practice, the sampling density $p$ defines the kernel we approximate, e.g., if we draw $\omega_i$ from the Gaussian distribution $\mathcal{N}(0, \ell^{-2}I_3)$ then $z(x)^\top z(y)$ approximates the squared exponential kernel $k(x, y) = \exp\left(-\|x - y\|^2/(2\ell^2)\right)$ with length scale $\ell > 0$. With mean random features $\widehat{\mu}_X := \frac{1}{m} \sum_{i=1}^{m} z(x_i)$ and $\widehat{\mu}_Y := \frac{1}{n} \sum_{j=1}^{n} z(y_j)$, (4) has the random-feature surrogate

$$\widehat{\text{MMD}}_{\text{RFF}}^2(X, Y) := \|\widehat{\mu}_X - \widehat{\mu}_Y\|_2^2. \tag{6}$$

Thus, we replace the quadratic kernel summations in (4) with linear-time feature means in (6). The number of sampled frequencies $D$ provides a direct trade-off. Because the frequencies are sampled randomly, the resulting surrogate is stochastic for finite $D$. Larger $D$ yields a more accurate kernel approximation, while smaller $D$ improves runtime and memory, enabling registration on large point clouds.

## 3. Our Approach

We now present MMD-Reg, which formulates rigid point-cloud registration as the minimization of a random-feature surrogate of the maximum mean discrepancy. Building on this formulation, we introduce Neural MMD-Reg, which leverages implicit differentiation to integrate MMD-Reg as a differentiable optimization layer in learning-based pipelines. We further describe a simple and effective network architecture for this pipeline based on set transformers.

**MMD-Reg.** In the registration setting, we evaluate (6) after applying a rigid transformation to the source points. Let $T_\theta(x) = R_\theta x + t_\theta$ denote a smoothly parametrized rigid transformation with parameters $\theta$ (e.g., a rotation and translation parametrization in $SE(3)$), and define the transformed source set $T_\theta(X) = \{T_\theta(x_i)\}_{i=1}^{m}$. We then define the objective function $F$, based on (6), as

$$F(\theta) := \widehat{\text{MMD}}_{\text{RFF}}^2(T_\theta(X), Y) = \|\widehat{\mu}_{T_\theta(X)} - \widehat{\mu}_Y\|_2^2. \tag{7}$$

This random-feature objective can be written directly as the nonlinear least-squares problem

$$F(\theta) = \|r(\theta)\|_2^2 = \sum_{k=1}^{2D} r_k(\theta)^2, \tag{8}$$

where $r(\theta) := \widehat{\mu}_{T_\theta(X)} - \widehat{\mu}_Y \in \mathbb{R}^{2D}$ has components

$$r_k(\theta) = \frac{1}{m} \sum_{i=1}^{m} z_k\big(T_\theta(x_i)\big) - \frac{1}{n} \sum_{j=1}^{n} z_k(y_j), \quad (9)$$

for $k = 1, \ldots, 2D$. Because the residuals depend on $\theta$ through the rotation $R_\theta$ and the trigonometric functions of the random-feature map, the objective is nonconvex. However, because it fits the standard nonlinear least-squares form, we can apply standard local solvers, such as Levenberg–Marquardt. Our aim is therefore not to provide new global convergence guarantees, but to obtain a practical correspondence-free objective that is effective in the regimes studied here. For clarity, we summarize the resulting MMD-Reg method in Algorithm 1. The estimated rigid transformation is given by $T_{\theta^\star}$.

When applying Algorithm 1, an effective choice of frequencies is

$$\omega_1/\ell, \ldots, \omega_D/\ell, \quad (10)$$

where $\ell > 0$ is a chosen scale parameter, and $\omega_1, \ldots, \omega_D$ are vectors in $\mathbb{R}^3$ whose components are sampled i.i.d. from either the standard Gaussian distribution $\mathcal{N}(0, 1)$ or the Laplace distribution $\mathrm{Laplace}(0, 1)$. In (5), we have $(\omega_i/\ell)^\top x = \omega_i^\top(x/\ell)$. Thus, small values of $\ell$ induce higher-frequency random features and increase sensitivity to fine-scale geometric structure, while large values emphasize broader alignment. Equivalently, scaling by $\ell$ corresponds to selecting the scale parameter of the stationary kernel approximated by $z(x)^\top z(y)$. Therefore, when the initial alignment is poor, Algorithm 1 can be applied sequentially with decreasing values of $\ell$. In this case, $\theta^\star$ is used as $\theta_0$ in the subsequent run, and a new sample of frequencies is drawn using a smaller scale parameter. We view this as a practical coarse-to-fine heuristic rather than an automatically selected or uniquely optimal schedule.

An important property of Algorithm 1 is that the solution $\theta^\star$ can be differentiated with respect to the input point clouds $X$ and $Y$ using IFT. Rather than deriving and implementing these derivatives by hand, we rely on the automatic implicit differentiation framework provided by JAXopt (Blondel et al., 2022). This allows us to efficiently compute gradients of $\theta^\star$ with respect to $X$ and $Y$ by solving linear systems, rather than unrolling the optimization algorithm and differentiating through its iterations. In practice, we find that JAXopt's Levenberg–Marquardt solver with an initial damping parameter of 1.0 provides robust convergence, and we adopt this setting for all experiments in Section 4. We do not provide a new convergence theorem for Levenberg–Marquardt under the random-feature approximation. Rather, the stability evidence in this paper is empirical. We additionally use JAX's vmap to batch the MMD-Reg layer during Neural MMD-Reg training.

---

**Algorithm 1** MMD-Reg

1: **Input:** Source point cloud $X = \{x_i\}_{i=1}^{m}$, target point cloud $Y = \{y_j\}_{j=1}^{n}$, random Fourier frequencies $\omega_1, \ldots, \omega_D$ as defined in (5), and initial transformation parameters $\theta_0$.
2: Find a solution $\theta^\star$ to the problem (8) using a standard nonlinear least-squares solver, such as Levenberg–Marquardt, initialized at $\theta_0$.
3: **Output:** Estimated transformation parameters $\theta^\star$.

---

**Neural MMD-Reg with unsupervised training.** Differentiability of the solution $\theta^\star$ with respect to the input point clouds enables an unsupervised training pipeline in which a neural network learns an initialization that aligns the inputs under the MMD objective with a chosen scale parameter $\ell$ as defined in (10). A network Net predicts an initial transformation $\theta_{\mathrm{Net}} = \mathrm{Net}(X, Y)$, which is applied to the source point cloud before running MMD-Reg from Algorithm 1 to compute a refined solution $\theta^\star$. The network is trained using a self-supervised loss encouraging $T_{\theta^\star}$ to be close to the identity, with gradients backpropagated through the MMD-Reg layer via implicit differentiation. At inference time, we apply the composed transformation $T_{\theta^\star} \circ T_{\theta_{\mathrm{Net}}}$, combining the learned initializer with the registration layer. This pipeline is most effective when $\ell$ is well matched to the data and the point clouds exhibit sufficient overlap.

**Neural MMD-Reg with supervised training.** When ground-truth transformations are available, the network can be trained in a supervised fashion to learn auxiliary parameters of MMD-Reg. Using the frequencies in (10), the network Net predicts an adaptive length scale $\ell_{\mathrm{Net}}$ together with an initial transformation $\theta_{\mathrm{Net}}$. MMD-Reg from Algorithm 1 is then applied to the input point clouds using the scaled frequencies $\omega_i/\ell_{\mathrm{Net}}$, initialized at $\theta_{\mathrm{Net}}$, to compute a refined solution $\theta^\star$. Supervision is applied to the network-predicted initialization using ground-truth transformations, while gradients with respect to $\ell_{\mathrm{Net}}$ are obtained by implicitly differentiating $\theta^\star$ through the MMD-Reg layer. At inference time, we run MMD-Reg initialized at $\theta_{\mathrm{Net}}$ using the learned length scale and return $T_{\theta^\star}$ as the final registration. In partial-overlap settings, the network additionally predicts pointwise overlap weights by producing logits $v \in \mathbb{R}^m$ and $w \in \mathbb{R}^n$ for $X$ and $Y$, respectively. The MMD-Reg layer then solves (8) with weighted residual components

$$r_k(\theta) = \frac{1}{m} \sum_{i=1}^{m} \sigma(v_i)\, z_k\big(T_\theta(x_i)\big) - \frac{1}{n} \sum_{j=1}^{n} \sigma(w_j)\, z_k(y_j),$$

where $\sigma$ denotes the sigmoid function. These weights are pointwise overlap weights on the input points, not learned weights on individual random Fourier features. These

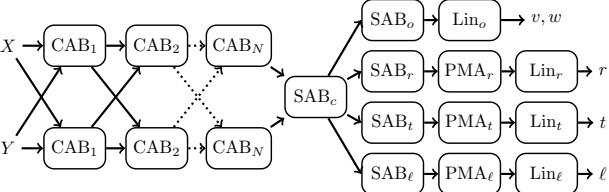

*Figure 1.* Set transformer architecture for Neural MMD-Reg. Alternating CABs process source and target point clouds, followed by concatenation and set attention. PMAs predicts global parameters, with the scale parameter constrained via a softplus nonlinearity.

weights are trained using a cross-entropy loss with ground-truth overlap masks, and gradients with respect to $\ell_{\text{Net}}$ are again obtained via implicit differentiation. The architecture of Net is illustrated in Figure 1 and described below.

**Set transformer architecture.** For Neural MMD-Reg, we use a set transformer architecture based on the attention blocks of (Lee et al., 2019), augmented with a cross-attention mechanism for paired point clouds. The model is composed of multihead attention blocks (MAB), set attention blocks (SAB), and pooling by multihead attention (PMA), with permutation invariance introduced through PMA. To enable interaction between the source and target point clouds, we construct a cross-attention block (CAB), which has the same structure as an MAB but takes queries from one set and keys and values from the other. We stack CABs in an alternating fashion, repeatedly updating features for each cloud conditioned on the other, without enforcing explicit correspondences. After cross-attention, features from both clouds are combined and passed through an SAB to form a fused representation. In the supervised setting, $\theta_{\text{Net}}$ and $\ell_{\text{Net}}$ are predicted using PMAs, while $v$ and $w$ are produced by an SAB. In the unsupervised setting, the network predicts only $\theta_{\text{Net}}$, and we switch to the induced variants of the attention blocks to enable linear-time self-attention computation in the number of points.

# 4. Experimental Results

In Section 4.1, we begin with controlled synthetic benchmarks for standalone local registration, disentangling the effects of noise, sampling density, and outliers while comparing MMD-Reg against widely used geometric and probabilistic baselines. We then move to large-scale outdoor LiDAR scans in Section 4.2 to assess standalone coarse-to-fine registration in outdoor environments with real sensors. These real outdoor datasets complement the synthetic benchmarks by evaluating registration across different geometric structure and sensing characteristics. Finally, in Section 4.3, we evaluate Neural MMD-Reg on standard learning-based benchmarks to examine the same registration objective when deployed as a differentiable refinement layer within end-to-end trainable models. Rotation and translation errors are reported as the average angular error in degrees (RRE) and Euclidean translation error (RTE), respectively.

## 4.1. PCPNet Dataset

We evaluate MMD-Reg on point clouds derived from the PCPNet dataset (Guerrero et al., 2018), which provides a controlled testbed for assessing robustness to noise and non-uniform sampling. Each object is provided as a densely sampled point cloud with 100 000 points, which we first center and normalize to the unit sphere. Source–target pairs are then constructed by uniformly subsampling two mutually exclusive point sets without replacement, thus avoiding trivial correspondences while preserving underlying geometry. A random rigid transformation is applied to the source, with Euler-angle rotations sampled uniformly from $[0°, 45°]$ and translations from $[-0.5, 0.5]$ independently per axis. We exclude shapes exhibiting rotational symmetries.

We compare MMD-Reg against well-established and widely used implementations of classical registration algorithms, ensuring that observed performance differences reflect the underlying formulations rather than artifacts of custom code. Geometric baselines from Open3D (Zhou et al., 2018) include ICP with point-to-point (ICP Pt2Pt) (Besl & McKay, 1992) and point-to-plane (ICP Pt2Pl) (Chen & Medioni, 1992) objectives, evaluated on both CPU and GPU, as well as Generalized ICP (GICP) (Segal et al., 2009). We further compare against probabilistic methods with reasonable runtime, namely FilterReg (Gao & Tedrake, 2019), GMMReg (Jian & Vemuri, 2011), and Support Vector Registration (SVR) (Campbell & Petersson, 2015), evaluated on CPU using the `probreg` library (Kenta-Tanaka et al.). We also report the runtime of Coherent Point Drift (CPD) (Myronenko & Song, 2010), from this library.

All methods use identical data generation and preprocessing, are initialized from the identity transformation, and run with fixed parameters across experiments. This setup is intended to evaluate local registration from a common initialization under controlled perturbations, rather than global convergence from arbitrarily large pose offsets. This ensures that comparisons isolate the behavior of the underlying registration objectives rather than differences arising from per-scenario hyperparameter tuning. For MMD-Reg, we use the frequencies in (10) with $\ell = 0.75$ and $D = 32$, and report results for both Gaussian (MMD-Reg G) and Laplace (MMD-Reg L) samples. In Table 1, we also report values of $D$ other than $D = 32$, which are indicated explicitly in the method name. We use a maximum correspondence distance of 0.75 for ICP and GICP, 100 mixture components for GMMReg, set the SVR parameter $\nu = 0.01$, and give CPD a maximum of 10 iterations. All other parameters are left at their default values.

*Table 1.* Registration errors (RRE↓, RTE↓) and runtime↓ on PCPNet data under High Noise, Gradient Density, and Striped Density settings (CPU/GPU). For the CPU results, the MMD-Reg $G_D$ rows illustrate the accuracy–runtime trade-off across feature dimensions: $D = 16$ is fastest but less accurate, $D = 32$ provides a strong balance, and $D = 64$ yields only marginal or mixed gains at higher cost. Across all three conditions, MMD-Reg (G/L) achieves the lowest errors overall. Relative to the non-MMD baselines in the table, the $D = 32$ MMD-Reg variants reduce RRE by **87–97%** and RTE by **83–96%**, while also reducing runtime by **18–90%** on CPU and **90–97%** on GPU. Per column (separately for CPU and GPU), the best value(s) are bold and the second-best value(s) are underlined.

| | High Noise | | | Gradient Density | | | Striped Density | | |
|---|---|---|---|---|---|---|---|---|---|
| Method | RRE (°) | RTE (-) | Time (ms) | RRE (°) | RTE (-) | Time (ms) | RRE (°) | RTE (-) | Time (ms) |
| **CPU** | | | | | | | | | |
| ICP Pt2Pt | 14.787 | 0.1066 | 1302 | 18.371 | 0.1548 | 2333 | 8.232 | 0.0536 | 1978 |
| ICP Pt2Pl | 6.246 | 0.0399 | 1378 | 15.465 | 0.1207 | 1374 | 8.754 | 0.0434 | 1005 |
| GICP | 6.731 | 0.0451 | 2705 | 19.411 | 0.1590 | 2000 | 11.740 | 0.0628 | 1555 |
| FilterReg | 9.472 | 0.0664 | 6720 | 13.016 | 0.0977 | 6744 | 5.752 | 0.0512 | 6700 |
| GMMReg | 19.746 | 0.1322 | 2209 | 20.114 | 0.1383 | 2781 | 18.852 | 0.1285 | 2507 |
| SVR | 15.080 | 0.1368 | 5113 | 16.570 | 0.1429 | 6477 | 15.092 | 0.1426 | 4373 |
| MMD-Reg $G_{16}$ (Ours) | 2.673 | 0.0193 | **390** | 2.062 | 0.0131 | **431** | 1.721 | 0.0110 | **385** |
| MMD-Reg $G_{32}$ (Ours) | 0.811 | 0.0068 | 684 | **0.886** | **0.0064** | 843 | 0.690 | 0.0064 | 824 |
| MMD-Reg $G_{64}$ (Ours) | 0.787 | 0.0066 | 1390 | 0.916 | 0.0066 | 1550 | 0.704 | 0.0064 | 1385 |
| MMD-Reg $L_{32}$ (Ours) | **0.716** | **0.0063** | 706 | 0.889 | **0.0064** | 833 | **0.631** | **0.0058** | 801 |
| **GPU** | | | | | | | | | |
| ICP Pt2Pt | 14.787 | 0.1066 | 555 | 18.371 | 0.1548 | 606 | 8.232 | 0.0535 | 576 |
| ICP Pt2Pl | 6.239 | 0.0399 | 535 | 15.420 | 0.1204 | 315 | 8.778 | 0.0436 | 225 |
| MMD-Reg $G_{32}$ (Ours) | 0.811 | 0.0068 | 20 | **0.886** | **0.0064** | 27 | 0.690 | **0.0064** | 22 |
| MMD-Reg $L_{32}$ (Ours) | **0.716** | **0.0063** | **19** | 0.888 | **0.0064** | **24** | **0.631** | **0.0058** | **20** |

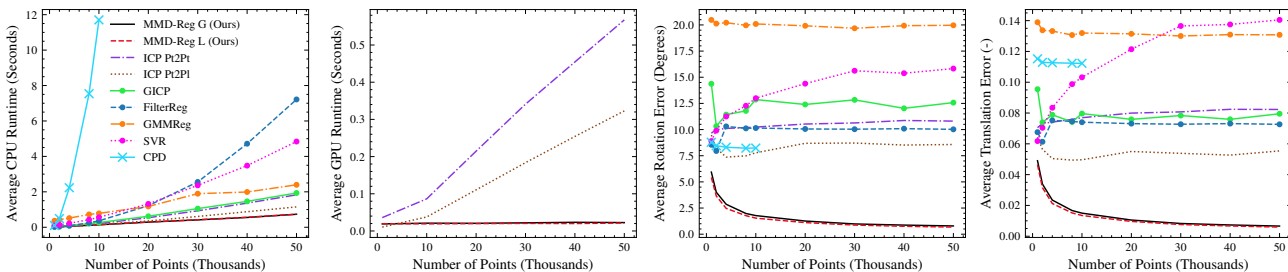

*Figure 2.* Average rotation error↓, translation error↓, and CPU/GPU runtime↓ versus the number of points on PCPNet. MMD-Reg shows gradual CPU scaling and near-constant GPU runtime with decreasing error, while correspondence-based (ICP, GICP) and probabilistic (FilterReg, GMMReg, SVR, CPD) baselines exhibit increasing runtimes with limited accuracy gains.

In Table 1, we consider three challenging conditions. The high-noise setting applies strong additive Gaussian perturbations to point coordinates, degrading local geometric structure and challenging methods that rely on precise neighborhood information. The gradient density setting introduces smoothly varying sampling density across the surface, resulting in systematic over- and under-sampling to test sensitivity to density imbalance. Finally, the striped density setting imposes abrupt, banded changes in sampling density, creating extreme non-uniformity with sharp transitions. Each setting contains seven asymmetric objects from the corresponding PCPNet test split, yielding 1400 source–target pairs per setting by processing each object 200 times, with each source and target containing 50 000 points. Registration examples from the gradient density and high-noise settings, as well as the 20% outlier setting discussed later, are shown in Figure 3.

In Figure 2, we examine how runtime and accuracy scale with the number of points. These experiments are conducted on the 42 asymmetric objects from the PCPNet test_all split, where each object is processed 30 times to generate 1260 source–target pairs per number of points. MMD-Reg exhibits a gradual increase in runtime on CPU and a near-constant runtime on GPU, while continuing to reduce error as the number of points increases. In contrast, classical correspondence-based methods based on ICP show steadily increasing runtimes with little improvement in accuracy, and probabilistic baselines incur substantially higher computational costs. In particular, CPD scales poorly with the number of points, becoming prohibitively slow beyond a few thousand points. These results highlight the favorable scalability of MMD-Reg on parallel hardware and its ability to leverage increasing point density for improved alignment.

In Figure 4, we evaluate robustness to outliers by increasing

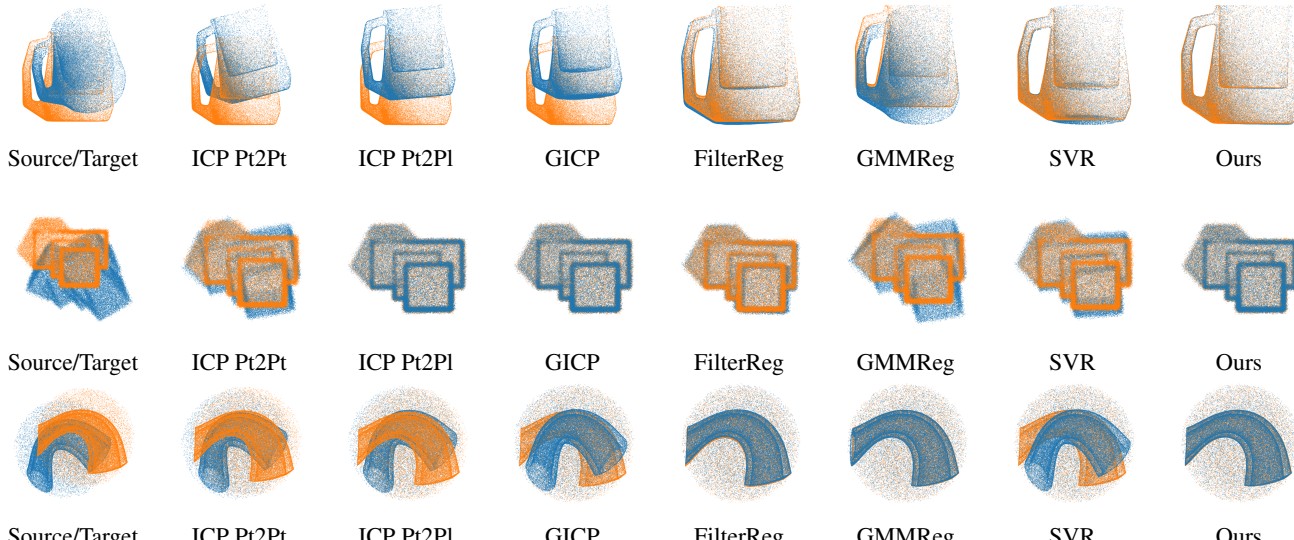

*Figure 3.* Registration examples on PCPNet under gradient density (top row), high noise (middle row), and 20% outlier (bottom row) settings. MMD-Reg (rightmost column) maintains accurate alignment across all conditions, ICP and GICP exhibit degraded performance in the presence of outliers when not augmented with robustness mechanisms.

the fraction of points in both source and target clouds that are replaced by uniformly sampled outliers from inside the unit ball prior to applying the random transformation. These experiments are conducted on the seven asymmetric objects from the PCPNet `test_no_noise` split, with each object processed 10 times to generate 70 source–target pairs per outlier percentage. MMD-Reg degrades gracefully as the outlier ratio increases, maintaining low error over a wide range. In contrast, correspondence-based methods ICP and GICP fail rapidly. This behavior is expected for methods that rely on hard nearest-neighbor correspondences unless augmented with robust loss functions or explicit outlier handling. Our goal here is to illustrate that MMD-Reg exhibits strong robustness to outliers without additional augmentation. Probabilistic baselines show mixed behavior.

### 4.2. Real Outdoor

We further evaluate our method on the Wild Places dataset (Knights et al., 2023), which consists of large-scale LiDAR scans collected in outdoor, unstructured natural environments, and on KITTI (Geiger et al., 2012), which contains driving scenes. Together, these datasets assess registration across two real outdoor regimes with different geometry and sensing characteristics. Wild Places contains sparse, irregular geometry, making it a challenging testbed for registration beyond structured urban scenes. All methods are evaluated on these datasets without additional sequence-specific preprocessing or tuning, so that observed performance differences more directly reflect the underlying registration objectives rather than differences in data preparation.

For Wild Places, we construct source–target point-cloud

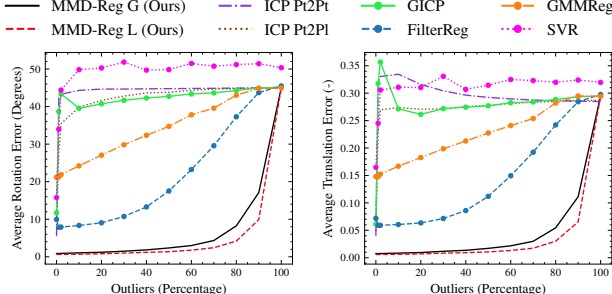

*Figure 4.* Average rotation error↓ and translation error↓ versus outlier percentage on PCPNet. MMD-Reg exhibits greater robustness to increasing outlier ratios than competing approaches.

pairs from the sequences K-03, K-04, V-03, and V-04. Pairs are formed by selecting frames separated by a stride of three, with the earlier and later frames treated as the source and target point clouds, respectively. All point clouds are voxel-downsampled with a voxel size of $0.15\,\mathrm{m}$ and then sampled to $50\,000$ points. The ground-truth rigid transformation is formed by computing the relative transformation between the poses associated with the source and target frames. For KITTI, we evaluate the same real outdoor registration setup on sequences 07, 08, 09, and 10 using the same voxel downsampling and method parameters. The only differences are that source–target pairs are formed using a stride of one, and after voxel downsampling we randomly sample $30\,000$ points per scan rather than $50\,000$.

In Tables 2 and 3, we compare MMD-Reg against geometric baselines on GPU. The intent of these experiments is to evaluate standalone local registration under a shared coarse-

*Table 2.* Registration errors (RRE↓, RTE↓) and runtime↓ on Wild Places sequences V-03, V-04, K-03, and K-04. MMD-Reg G/L denote our method with Gaussian/Laplace random frequencies. Relative to the multi-scale ICP baselines, across all four sequences, MMD-Reg (G/L) reduces RRE by **61–85%**, RTE by **40–77%**, and runtime by **6–50%**. Per column, best is bold and second-best underlined.

| Method | Sequence V-03 | | | Sequence V-04 | | | Sequence K-03 | | | Sequence K-04 | | |
| --- | --- | --- | --- | --- | --- | --- | --- | --- | --- | --- | --- | --- |
| | RRE (°) | RTE (m) | Time (s) | RRE (°) | RTE (m) | Time (s) | RRE (°) | RTE (m) | Time (s) | RRE (°) | RTE (m) | Time (s) |
| MS ICP Pt2Pt | 1.484 | 0.092 | 0.197 | 1.264 | 0.084 | 0.202 | 1.743 | 0.151 | 0.204 | 1.178 | 0.124 | 0.202 |
| MS ICP Pt2Pl | 1.229 | 0.085 | 0.140 | 0.970 | 0.073 | 0.142 | 1.305 | 0.123 | 0.145 | 0.742 | 0.081 | 0.141 |
| MMD-Reg G | 0.410 | 0.045 | **0.107** | 0.383 | 0.044 | **0.109** | **0.452** | **0.046** | **0.102** | **0.173** | **0.029** | **0.105** |
| MMD-Reg L | **0.405** | **0.040** | 0.131 | **0.327** | **0.038** | 0.127 | 0.501 | 0.061 | 0.115 | 0.179 | 0.031 | 0.126 |

*Table 3.* Registration errors (RRE↓, RTE↓) and runtime↓ on KITTI sequences 07, 08, 09, and 10. Compared with the multi-scale ICP baselines, MMD-Reg is competitive but does not outperform point-to-plane ICP on these driving sequences. Per column, the best value is bold and the second-best value is underlined.

| Method | Sequence 07 | | | Sequence 08 | | | Sequence 09 | | | Sequence 10 | | |
| --- | --- | --- | --- | --- | --- | --- | --- | --- | --- | --- | --- | --- |
| | RRE (°) | RTE (m) | Time (s) | RRE (°) | RTE (m) | Time (s) | RRE (°) | RTE (m) | Time (s) | RRE (°) | RTE (m) | Time (s) |
| MS ICP Pt2Pt | 0.081 | 0.018 | 0.092 | 0.084 | 0.026 | 0.101 | 0.077 | 0.017 | 0.112 | 0.094 | 0.019 | 0.103 |
| MS ICP Pt2Pl | **0.049** | **0.012** | **0.040** | **0.060** | **0.024** | **0.044** | **0.053** | **0.016** | **0.046** | **0.066** | **0.014** | **0.042** |
| MMD-Reg G | 0.081 | 0.016 | 0.082 | 0.092 | 0.028 | 0.077 | 0.093 | 0.023 | 0.085 | 0.099 | 0.019 | 0.081 |
| MMD-Reg L | 0.073 | 0.015 | 0.087 | 0.085 | 0.027 | 0.084 | 0.088 | 0.021 | 0.093 | 0.090 | 0.018 | 0.087 |

to-fine protocol on large outdoor scans, which is why we compare against multi-scale ICP variants operating in the same regime. Future work will broaden these comparisons to additional robust and learning-based methods, as well as to a wider range of real-world datasets.

Here, we consider multi-stage registration frameworks, which are essential for robust alignment on these real outdoor datasets given their large pose offsets, sparse geometry, and large spatial scale variation. Within this framework, we evaluate multi-scale ICP from Open3D (Zhou et al., 2018), considering both point-to-point (MS ICP Pt2Pt) (Besl & McKay, 1992) and point-to-plane (MS ICP Pt2Pl) (Chen & Medioni, 1992) variants. Registration is initialized from the identity transformation and proceeds through five fixed scales with voxel sizes $\{0.35, 0.30, 0.25, 0.20, 0.15\}$ m and corresponding maximum correspondence distances $\{4.0, 2.0, 1.0, 0.5, 0.25\}$ m. We also evaluate MMD-Reg in a sequential configuration, as described in Section 3. Our method is applied sequentially using five sets of random Fourier features with dimensions $D = \{256, 256, 256, 256, 512\}$ and decreasing kernel scales $\ell = \{4.0, 2.0, 1.0, 0.5, 0.25\}$, beginning from the identity transformation and initializing each stage from the previous solution. The decreasing $\ell$-schedule is a practical coarse-to-fine heuristic, where larger scales emphasize broader alignment and smaller scales refine finer geometric structure, and the larger final value of $D$ is used to improve approximation quality in the last refinement stage. This yields refinement analogous in spirit to multi-scale ICP, but operating on distributional discrepancies rather than explicit point correspondences. We report results for both Gaussian (MMD-Reg G) and Laplace (MMD-Reg L) random feature distributions.

### 4.3. ModelNet40

We evaluate Neural MMD-Reg in both supervised and unsupervised settings on the ModelNet40 dataset (Wu et al., 2015), a widely used benchmark for 3D shape analysis and point-cloud registration. ModelNet40 consists of CAD models from 40 object categories with diverse geometries, including objects exhibiting rotational symmetries. We generate source–target pairs by uniformly sampling point clouds with 1024 points from object meshes, normalizing them, and applying the same random rigid transformations as in Section 4.1 to the source cloud.

**Unsupervised Neural MMD-Reg (clean setting).** In the unsupervised setting, we consider a clean and fully overlapping registration scenario following the protocol of (Wang & Solomon, 2019), where no noise or cropping is applied. We retain all 40 object categories, including symmetric ones, and use identical samples for the source and target point clouds prior to the transformation, ensuring that a unique ground-truth alignment exists for each pair despite the presence of symmetry. This setting is challenging because symmetric configurations can induce competing transformations that yield similar distributional discrepancies, leading to shallow or ambiguous optimization landscapes.

In Table 4, we report results for unsupervised Neural MMD-Reg, as described in Section 3, using both Gaussian (Neural MMD-Reg G) and Laplace (Neural MMD-Reg L) random feature distributions. To isolate the contribution of the MMD-Reg refinement layer relative to the learned initializer, we additionally remove the MMD-Reg layer after training and report the registration accuracy obtained from the set transformer prediction alone, denoted by Set-Transformer G and Set-Transformer L. During training, the MMD-Reg in-

*Table 4.* ModelNet40 clean registration results. Per column the best value is shown in bold, the second-best is underlined and ties are marked with both. Results for methods other than ours are taken from (Peng et al., 2024; Chen et al., 2025).

| Method | RRE (°) | RTE (-) |
|---|---|---|
| ICP (Besl & McKay, 1992) | 6.407 | 0.0506 |
| Go-ICP (Yang et al., 2016) | 6.749 | 0.0109 |
| FGR (Zhou et al., 2016) | 0.022 | 0.0002 |
| PointNet-LK (Aoki et al., 2019) | 0.847 | 0.0054 |
| DCP-v2 (Wang & Solomon, 2019) | 3.992 | 0.0292 |
| RPM-Net (Yew & Lee, 2020) | 0.056 | 0.0003 |
| REGTR (Yew & Lee, 2022) | 1.186 | 0.0098 |
| GeoTransformer (Qin et al., 2023) | 0.634 | 0.0065 |
| DCMR (Peng et al., 2024) | 0.533 | 0.0064 |
| RPMNet++ (Chen et al., 2025) | 0.030 | 0.0002 |
| Set-Transformer G (Ours) | 0.235 | 0.0069 |
| Set-Transformer L (Ours) | 0.257 | 0.0122 |
| Neural MMD-Reg G (Ours) | 0.008 | <**0.0001** |
| Neural MMD-Reg L (Ours) | **0.006** | <**0.0001** |

*Table 5.* ModelNet40 partial-overlap registration results. Per column the best value is shown in bold, the second-best is underlined. Results for methods other than ours are taken from (Cheng et al., 2025; Cao et al., 2025).

| Method | RRE (°) | RTE (-) |
|---|---|---|
| RPM-Net (Yew & Lee, 2020) | 2.357 | 0.028 |
| RGM (Fu et al., 2021) | 4.548 | 0.049 |
| Predator (Huang et al., 2021) | 2.064 | 0.023 |
| CoFiNet (Yu et al., 2021) | 3.584 | 0.044 |
| GeoTransformer (Qin et al., 2023) | 2.160 | 0.024 |
| RID-Net (Cheng et al., 2025) | 1.695 | 0.019 |
| DMS (Cao et al., 2025) | 1.319 | 0.015 |
| Set-Transformer (Ours) | 1.577 | 0.022 |
| Neural MMD-Reg (Ours) | **1.292** | **0.014** |

ner problem is solved using a relatively small number of random Fourier features ($D = 32$), with frequencies resampled at each iteration. At test time, MMD-Reg is applied using a larger ($D = 128$) set of random features, yielding a more accurate approximation of the MMD objective. This separation encourages the network to learn an initializer that is robust to stochastic kernel approximations during training, rather than adapting to a single fixed random draw, while allowing high-accuracy registration at inference through a richer random-feature representation.

**Supervised Neural MMD-Reg (partial overlap setting).** For supervised Neural MMD-Reg, we consider the partially overlapping registration setting introduced by GeoTransformer (Qin et al., 2023), which removes eight symmetric categories to reduce ambiguity arising from rotational symmetries. This restriction is appropriate in this regime, where the source and target point clouds are independently sampled and corrupted with additive Gaussian noise. Partial overlap is simulated by cropping each point cloud using random hyperplanes, leaving 717 points per cloud.

In Table 5, we report results for supervised Neural MMD-Reg, as described in Section 3, using a Laplace random feature distribution. We also report the registration accuracy obtained from the set transformer prediction alone, without the MMD-Reg layer, to isolate the contribution of the refinement step. This baseline is denoted by Set-Transformer.

We perform model optimization in two distinct phases. In the first phase, the network is trained without the MMD-Reg layer to predict the relative rotation, translation, and pointwise overlap masks using supervised losses. During this phase, the kernel length-scale prediction head is present but not optimized, and no MMD optimization is performed. In the second phase, we restore the pretrained

network and freeze all parameters except the length-scale head. The model is then tuned by backpropagating through the weighted MMD-Reg optimization layer via implicit differentiation, using the network predictions as initialization and the predicted overlap masks to weight the residuals. In the second phase and at test time, we use a random-feature dimension of $D = 16384$, as larger values of $D$ appeared beneficial for this dataset, possibly due to the combination of noise and the small number of points per cloud (717). This staged procedure allows the network to first learn a stable geometric initialization and overlap estimation, and subsequently adapt the kernel scale to the downstream registration objective without destabilizing the learned representation.

## 5. Conclusion and Future Work

We presented MMD-Reg, a differentiable, correspondence-free method for rigid point-cloud registration that, for fixed random-feature dimension, scales linearly with point-cloud size. In the evaluated synthetic and real outdoor settings, it combines strong accuracy with fast inference against the compared baselines. The present paper is intended as a practical local registration objective and differentiable optimization layer, rather than a globally convergent solver for all initialization regimes. We further demonstrated accurate registration in unsupervised and supervised settings with full and partial overlap by integrating MMD-Reg as a differentiable optimization layer within neural networks. Future work will explore the integration of MMD-Reg into SLAM systems to leverage its robustness, efficiency, and accuracy. Neural MMD-Reg is not limited to the set transformer used here, and future work will explore alternative neural architectures. While our formulation currently operates on spatial point distributions, it can be extended to incorporate additional per-point attributes, such as semantic labels, learned feature embeddings, or color, via additional kernel components.

## Acknowledgements

This project was supported by resources and expertise provided by CSIRO IMT Scientific Computing.

## Impact Statement

This paper presents work whose goal is to advance point-cloud registration and the field of Machine Learning. There are many potential societal consequences of our work, none which we feel must be specifically highlighted here.

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
