# OpenReview forum: "Scalable and Differentiable Point-Cloud Registration Using Maximum Mean Discrepancy"
_ICML.cc/2026/Conference — ICML 2026 regular_

### Official Review · Reviewer_sJm6 · 2026-03-11

**Soundness:** 2
**Presentation:** 3
**Significance:** 3
**Originality:** 4
**Overall Recommendation:** 4
**Confidence:** 4

**Summary:**

This paper introduces MMD-Reg, a novel correspondence-free approach for point cloud registration. Instead of relying on point-to-point matching like traditional geometric methods, it treats the two point clouds as sample sets from spatial probability distributions and minimizes the Maximum Mean Discrepancy (MMD) between them. To achieve scalability, the authors utilize Random Fourier Features (RFF) to approximate the MMD kernel. The resulting formulation is a nonlinear least-squares objective. By combining MMD with RFF, the method achieves robust alignment against noise, varying sampling densities, and outliers without requiring explicit filtering or hard correspondences .

**Compliance With Llm Reviewing Policy:**

Affirmed.

**Final Justification:**

Thanks for your rebuttal. I will improve my original score.

**Key Questions For Authors:**

1.Could the authors provide experimental results, or at least a detailed discussion, on how Neural MMD-Reg and standalone MMD-Reg perform on standard, widely adopted benchmarks such as 3DMatch (for complex indoor scenes with severe occlusions) and the KITTI Odometry dataset ?

2.How does the stochastic variance of the RFF approximation impact the stability of the Levenberg-Marquardt optimizer and the overall training dynamics of Neural MMD-Reg? Can the authors provide empirical evidence or theoretical bounds showing that this variance does not prevent convergence to a high-quality local minimum, especially when D is small?

3.In the Wild Places experiment, the standalone MMD-Reg is applied sequentially using a very specific set of decreasing kernel scales l={4.0,2.0,1.0,0.5,0.25} and feature dimensions D={256,256,256,256,512}. What are the guiding principles, heuristics, or automated methods for selecting the sequence of kernel scales (l) and feature dimensions (D) when applying this sequential MMD-Reg to a completely new, unseen dataset? Is the method highly sensitive to these specific values?

**Limitations:**

Please refer to weakness and question.

**Strengths And Weaknesses:**

Soundness：
Strengths：The core technical approach is rigorously formulated. Approximating the Maximum Mean Discrepancy (MMD) with Random Fourier Features (RFF) to reduce the computational complexity from quadratic to linear is theoretically sound and well-justified.

Presentation：
The submission is very well-structured. The transition from the background of MMD to its empirical estimation, and finally to the proposed MMD-Reg algorithm, is highly logical and easy for an expert to follow.

Significance：
The paper addresses a highly relevant bottleneck in 3D vision: bridging the gap between scalable, robust geometric algorithms and end-to-end deep learning pipelines via a differentiable optimization layer .

Originality：
(1) It shifts the traditional registration paradigm away from finding brittle, explicit point-to-point correspondences, offering a fresh perspective on how to align geometries purely based on continuous distributional discrepancies.
(2) The core building blocks themselves—MMD, Random Fourier Features, the Implicit Function Theorem, and Set Transformers—are all well-established in the machine learning literature. The theoretical contribution is modest; the true novelty lies primarily in their systematic, engineering-level application to the specific domain of 3D vision.

Weaknesses：

(1)The method relies on the random sampling of Fourier frequencies, using D=32 features during training and D=128 during inference. However, there is a lack of deep theoretical analysis regarding the variance of this stochastic objective function and how it affects the guarantee of convergence.

(2)While evaluated on ModelNet40, PCPNet, and Wild Places, the paper completely lacks evaluation on standard, highly complex indoor benchmarks (like 3DMatch and 3DLoMatch) or standard autonomous driving drift benchmarks (like KITTI), which are typical for proving true real-world robustness in registration.

---

> ### Author Rebuttal · Authors · 2026-03-31
>
> We thank the reviewer for the careful reading and for the positive assessment of the formulation, presentation, and significance of the work. We are encouraged that the reviewer found the progression from the MMD background to the final algorithm clear, and that the correspondence-free formulation was viewed as a meaningful contribution.
>
> Regarding the concern about stochastic variance from the RFF approximation, we agree that this is an important aspect of the method. In the current paper, we selected $D$ to balance approximation quality and runtime, rather than aggressively tuning it for each experiment. Empirically, we have found MMD-Reg to be fairly robust to the choice of $D$. We agree, however, that this would be clearer with an explicit ablation, and we will add such results in the revision to better illustrate the trade-off between accuracy, runtime, and optimization stability.
>
> As a preliminary illustration, we include below a small CPU example on PCPNet showing the effect of varying $D$. These results are intended only as an initial indication for the reviewer. In the revision, we will provide a more comprehensive ablation and discussion.
>
> | Method           | RRE (°) | RTE (-) | Time (s) |
> |------------------|---------|---------|----------|
> | MMD-Reg G (D=16) | 2.673   | 0.0193  | 0.524    |
> | MMD-Reg G (D=32) | 0.811   | 0.0068  | 0.684    |
> | MMD-Reg G (D=64) | 0.787   | 0.0066  | 1.640    |
>
> We also note that Neural MMD-Reg already includes one mechanism that improves robustness to approximation error: the random frequencies are resampled across training iterations. This encourages the learned initializer to remain stable under stochastic kernel approximations rather than adapting to a single fixed random draw.
>
> More broadly, the paper does not claim a new convergence theorem for Levenberg--Marquardt under stochastic random-feature approximation, and we agree that this limitation should be stated more explicitly. Our present scope is empirical: we use the standard RFF approximation to obtain a linear-time surrogate, and then evaluate whether the resulting objective is sufficiently stable to support both standalone registration and differentiable training in the settings considered.
>
> We also agree that evaluation on additional standard registration benchmarks would strengthen the paper. We have already obtained encouraging preliminary results for standalone MMD-Reg on KITTI, and we include a small example below to give the reviewer an initial indication. In the revision, we will add a more comprehensive benchmark section and broader discussion to better position the method on standard registration testbeds.
>
> | Method         | RRE (°) | RTE (m) | Time (s) |
> |----------------|---------|---------|----------|
> | MS ICP Pt2Pt   | 0.110   | 0.021   | 0.096    |
> | MS ICP Pt2Pl   | 0.075   | 0.018   | 0.041    |
> | MMD-Reg G      | 0.110   | 0.021   | 0.087    |
> | MMD-Reg L      | 0.104   | 0.020   | 0.089    |
>
> We also appreciate the question of how to choose the sequence of kernel scales $\ell$ and feature dimensions $D$ for sequential MMD-Reg on a new dataset. In our formulation, $\ell$ controls the spatial scale of the random-feature approximation: larger values emphasize broader alignment, while smaller values increase sensitivity to finer geometric structure. When the initial alignment is poor, we therefore apply MMD-Reg sequentially with decreasing values of $\ell$, using the previous estimate to initialize the next stage. The schedule used on Wild Places is intended as a coarse-to-fine refinement strategy, analogous in spirit to multi-scale ICP.
>
> We will revise the text to emphasize that the Wild Places schedule is an instance of this general coarse-to-fine principle, rather than a claim that the specific values $\{4.0,2.0,1.0,0.5,0.25\}$ are uniquely optimal. Similarly, $D$ controls the approximation-quality/runtime trade-off of the RFF surrogate: larger $D$ improves the kernel approximation at higher computational cost, while smaller $D$ favors speed and memory efficiency. We therefore used moderate feature dimensions in the early Wild Places stages and a larger value in the final refinement stage. We will clarify in the revision that this is a practical heuristic rather than an automated selection rule.
>
> We also note that the current paper already includes one form of automatic adaptation in the learning-based setting. In supervised Neural MMD-Reg, the network predicts the kernel length scale $\ell_{\mathrm{Net}}$, providing a learned adaptation of the scale parameter rather than relying only on a fixed heuristic value.
>
> We thank the reviewer again for the constructive comments. The points raised are very helpful in clarifying where the empirical presentation can be strengthened without changing the core claims of the paper. We believe these revisions will make the paper's contribution and intended scope more clear.

---

> > ### Author Rebuttal · Reviewer_sJm6 · 2026-04-03
> >
> > Thanks for your rebuttal. I will improve my original score.

---

> > > ### Author Response · Authors · 2026-04-07
> > >
> > > Thank you again for the discussion and for letting us know that our rebuttal addressed your concerns. We appreciate the time you've taken with the paper. We just wanted to note that the review status visible to us still appears unchanged, in case this is simply a system or visibility issue. Thank you again for your consideration.

---

### Official Review · Reviewer_fyVJ · 2026-03-12

**Soundness:** 3
**Presentation:** 3
**Significance:** 2
**Originality:** 3
**Overall Recommendation:** 4
**Confidence:** 3

**Summary:**

The paper proposes a method for local (i.e., numerically optimization from a starting pose) rigid point cloud registration. As comparison measure, it compares correlations between density functions obtained with a kernel regressor. In my undertanding*) the method boils down to performing an approximate Fourier-transform of the point set by projecting the points on randomly chosen Fourier basis functions that are band-limited by the "length" of the direction vectors $l$. By changing $l$, one can tune the amount of spatial coarse-graining (band-limiting) applied to the geometry, allowing for a multi-resolution registration algorithms that starts with a coarse-scale alignment and subsequently adds details.
Alignment itself is performed by traditional numerical descent (Levenberg-Marquard), but the hyperparameters (band-limits, potentially anisotropic weights of the random Fourier features/directional frequencies) can be controlled by a neural network.
The overall method performs very well in comparison to various base-lines and prior work, including simple ICP-variants as well as band-limitation-based methods such as FilterReg. I would assume that control by the neural network is essential for these improvements, but I have not seen an explicit ablation study.


*) I am not an expert in kernel methods; so please correct me in the rebuttal if I misunderstood aspects of this work.

**Compliance With Llm Reviewing Policy:**

Affirmed.

**Final Justification:**

The rebuttal was very helpful in clarifying several technical details. In terms of the big picture, I would stick to my original rating: The method is technically interesting, but does not achieve something fundamentally novel - it is just implemented in a different way and understood from an interesting point of view. I would thus see this as a good contribution, but it does not "feel mandatory" to accept the paper at ICML. Thus, I am giving an overall positive, but not very strong recommendation.

**Key Questions For Authors:**

Technical: Could you replace the random-fourier projection by simpler (quasi) finite-impulse-response filters such as (truncated/approximated) Gaussians in the spatial domain and get similar results (maybe simpler/faster)? If so, what would be lost in such a simplification?

Evaluation/Ablation: How important is the control by the network and the individual weighting of Fourier-functions? (The latter being a decisive factor would emphasize the importance of the concrete technical route taken here, so this would strengthen the paper in my view; the first question is targeting the understanding of why it works so well; maybe this question could also be answered directly)

Are there applications where it is important to have a differentiable local registration method with stronger convergence properties (in other words, why not stick to the "standard" global+local approach such as a stereotypical RANSAC-variant plus robustified [e.g., truncated attraction] ICP variant)?

My opinion is already leaning positively at this point; a better motivation of the choices made would strengthen the importance/significance of the proposed approach.

**Limitations:**

It would be good to show results of real-world scans with partial acquisition / limited overlap in a figure (only quantitative results are reported) as this could be a weak point of the correlation through global local-pass filter functions, and in general a big challenge for local matching methods.
I do not see specific unaddressed social challenges.

**Strengths And Weaknesses:**

Strengths:
- The paper is well written, and reasonably easy to understand.
- The method is not too difficult to implement and seems to be practical.
- Results are very good, and comparisons include obvious base-lines in the form of similar prior work (Gaussian filters for band-limiting).
- The idea of using anisotropic weighting (weighting each Fourier-direction) is nice, as well as the automatic hyperparameter control by a network.

Weaknesses:
- Minor weakness/suggestion: The kernel formulation is a bit heavy and requires a solid foundation in this area of ML; one could motivate the method also more directly by formulating the method as approximate band-limiting of a sampled characteristic functions, or as a positional encoding.
- Originality/Novelty: The idea of band-limiting geometry by applying a low-pass-filter to the point set, and subsequently aligning at a chosen (and potentially adaptable) level of resolution is not new; many of the cited references include this idea; one of the first references that does this explicitly with Gaussians is the "normal distribution transform" by Biber and Straßer (IROS 2003).
- Costs: Random Fourier Features are still costly, as they compute an approximate (non-fast) Fourier transform. Using finite-impulse-response filters such as (slightly truncated) Gaussians and the like, in companion with band-limited subsampling of the point cloud could speed up the method quite a bit while achieving similar (maybe even better) results for isotropic weighting (for per-frequency weighting, this is hard to emulate).
- Evaluation: It would be great to have an ablation study that shows the impact of the network control, and of the various weighting schemes (e.g.: per frequency/direction, just the scale, fixed schedule, fixed resolution), and maybe of approximating the kernel.
- Evaluation/Significance: To be fully fair, one would have to compare against a global + local method, such a RANSAC-based prealignment followed by ICP with outlier-robust distances, which is the traditional "standard pipeline" and would certainly fare better in a direct comparison (one would rarely use a plain local method for global shape registration in practice when adequate efforts are made to solve the problem). One has to counter this point of course by also taking into account that this is a local optimization method, so while the problem does have already good solutions, the "full" global+local pipeline is also a bit out of scope. However, the potential impact is more limited, as good, easy and efficient practical solutions for robust global alignment do exist.

Overall, the idea is nice, but if my understanding is correct, it is not very novel on a conceptual level; just the implementation through random feature maps of kernels is a bit unconventional. Together with the automated control, the method seems to yield very good results, which is of course an argument in favor of the paper. My recommendation would thus be mildly positive, but not very strong due to not introducing very novel ideas and probably limited practical impact in the primary application domain of point cloud merging against the broader state-of-the-art.

---

> ### Author Rebuttal · Authors · 2026-03-31
>
> We thank the reviewer for the careful reading, constructive feedback, and positive assessment of the paper's clarity, practicality, and empirical performance. We are encouraged that the method was found reasonably easy to understand and implement, and that the comparisons with geometric and probabilistic baselines were seen as meaningful.
>
> First, we would like to clarify one important point of interpretation. The paper does not learn per-frequency/per-direction weights for the random Fourier features. Standalone MMD-Reg has no learned component at all, and the strong results on PCPNet and Wild Places are obtained entirely in this non-learning setting. In Neural MMD-Reg, the network predicts only an initial estimate of the rigid transformation in the unsupervised setting, and in the supervised partial-overlap setting it predicts an initial estimate, a kernel length scale, and pointwise overlap weights on the input points. Thus, the adaptive weighting in the paper is per-point, not per-frequency. This distinction is important because it shows that the core MMD-Reg objective is already effective on its own, while differentiability extends it to end-to-end trainable pipelines.
>
> We also appreciate the reviewer's spectral/filtering interpretation. We agree this view is valuable and can be made clearer in the paper. Our formulation uses random Fourier features to approximate a shift-invariant kernel, so that the MMD between the transformed source and target distributions becomes the squared Euclidean distance between their mean random features. By Bochner's theorem, this admits a natural spectral/multiresolution interpretation, with the kernel scale controlling coarse-to-fine behavior. We will make this connection more explicit in the revision.
>
> Regarding novelty, we agree that coarse-to-fine registration is not new in itself. Our contribution is more specific: (i) a correspondence-free MMD formulation of rigid registration, (ii) a random-feature surrogate with linear scaling in the number of points for fixed feature dimension, (iii) a standard nonlinear least-squares objective solvable by Levenberg--Marquardt, and (iv) implicit differentiation of the optimizer so it can be used as a differentiable layer in end-to-end trainable models. We will revise the introduction and related work to better distinguish this contribution from prior smoothing, filtering, and multiscale approaches.
>
> On the question of simpler spatial-domain filters (e.g., Gaussian/truncated-Gaussian), we agree these are natural alternatives for smoothing geometry before alignment. Our claim is not that RFFs are the only viable route, but that they provide a convenient linear-scaling kernel-discrepancy surrogate in terms of feature means, leading directly to a standard least-squares formulation and a straightforward path to implicit differentiation. Spatial filtering methods may be useful alternatives, but they follow a different approximation route and do not automatically retain the same mean-feature least-squares structure or the same kernel/MMD interpretation. With characteristic kernels, MMD is a metric on probability measures, which gives the objective a clear distributional meaning.
>
> Regarding ablations, the paper already compares Gaussian and Laplace random-feature distributions in several experiments. We will also include an ablation on the feature dimension $D$, which governs the approximation-quality/runtime trade-off of the RFF surrogate. In the learning-based experiments, the paper includes a partial ablation through the set-transformer-only baselines: on ModelNet40, we report both network-only accuracy and accuracy after applying the MMD-Reg layer, isolating the contribution of the optimization layer.
>
> Finally, we appreciate the broader question of why a differentiable local registration method is useful given strong global+local pipelines. We agree that such pipelines are important and effective, and our goal is not to replace them universally. Rather, we focus on a local registration objective that is correspondence-free, scalable, and differentiable with respect to the inputs. This is especially useful when registration is a module inside a learned system, where gradients through the refinement step are required, and also in settings where a coarse initialization is already available (such as navigation tasks).
>
> We thank the reviewer again for the helpful comments and positive overall recommendation. The points raised clearly identify where the explanation can be strengthened without changing the paper's core claims. We believe these revisions will make the contribution and intended use cases more transparent and strengthen the case for the method as a practical, differentiable registration approach.

---

> > ### Author Rebuttal · Reviewer_fyVJ · 2026-04-03
> >
> > Thanks for the comprehensive and friendly discussion; I would see all important aspects discussed/resolved.

---

### Official Review · Reviewer_rdAv · 2026-03-12

**Soundness:** 3
**Presentation:** 3
**Significance:** 2
**Originality:** 2
**Overall Recommendation:** 4
**Confidence:** 3

**Summary:**

The paper presents a scalable, differentiable method for point cloud registration based on MMD. It uses random Fourier features to approximate the kernel objective efficiently, making computation roughly linear in the number of points for fixed feature dimension. Experiments on synthetic and real datasets show competitive accuracy with improved scalability.

**Compliance With Llm Reviewing Policy:**

Affirmed.

**Final Justification:**

Thanks for the authors' response and addressed my concern, so I keep my positive scores.

**Key Questions For Authors:**

- Sensitivity to random features.
The method relies on random Fourier features to approximate the MMD objective. Could the authors provide a systematic analysis of how the number of random features affects registration accuracy, runtime, and convergence stability?

- Robustness to large initial misalignment.
In the PCPNet experiments, all methods appear to be initialized from the identity transformation with rotations limited to about 45^\circ. How does the method perform under larger initial rotations (e.g., 90^\circ or 180^\circ), where the non-convex objective may be more prone to local minima?

- Strength of the scalability claim.
The paper emphasizes scalability but the large-scale experiments appear to use downsampled point clouds. Could the authors provide results on larger raw point clouds or more detailed runtime and memory analyses to better support the scalability claim?

**Limitations:**

yes

**Strengths And Weaknesses:**

## Strengths

The paper addresses an important problem: making point cloud registration both scalable and differentiable.
Its core idea of combining MMD-based distribution alignment with random Fourier features is simple, elegant, and computationally appealing.

## Weaknesses

The paper does not provide a thorough ablation on the number of random features, so the trade-off between approximation quality, runtime, and convergence stability is not fully understood.
The evaluation under large initial rotations or severe misalignment is limited, which leaves some uncertainty about robustness given the non-convex objective.

---

> ### Author Rebuttal · Authors · 2026-03-31
>
> We thank the reviewer for the careful reading of the paper, for the thoughtful summary, and for the positive assessment of the core idea. We are encouraged that the reviewer found the combination of MMD-based distribution alignment and random Fourier features to be simple, elegant, and computationally appealing.
>
> Regarding sensitivity to the number of random features, we agree that a more systematic analysis would strengthen the paper. In the current paper, we selected $D$ to balance approximation quality and runtime, rather than aggressively tuning it for each experiment. Empirically, we have found MMD-Reg to be fairly robust to the choice of $D$. We agree, however, that this would be clearer with an explicit ablation, and we will add such results in the revision to better illustrate the trade-off between accuracy, runtime, and optimization stability.
>
> As a preliminary illustration, we include below a small CPU example on PCPNet showing the effect of varying $D$. These results are intended only as an initial indication for the reviewer. In the revision, we will provide a more comprehensive ablation and discussion.
>
> | Method           | RRE (°) | RTE (-) | Time (s) |
> |------------------|---------|---------|----------|
> | MMD-Reg G (D=16) | 2.673   | 0.0193  | 0.524    |
> | MMD-Reg G (D=32) | 0.811   | 0.0068  | 0.684    |
> | MMD-Reg G (D=64) | 0.787   | 0.0066  | 1.640    |
>
> We also note that Neural MMD-Reg already includes one mechanism that improves robustness to approximation error: the random frequencies are resampled across training iterations. This encourages the learned initializer to remain stable under stochastic kernel approximations rather than adapting to a single fixed random draw.
>
> On robustness to larger initial misalignment, we agree that the PCPNet setup is a local-registration regime, where all methods are initialized from the identity transformation. This was intended to isolate robustness to noise, density imbalance, and outliers in a controlled setting rather than to study very large pose offsets. At the same time, the paper does address the large-misalignment regime in the Wild Places experiments. In our formulation, $\ell$ controls the spatial scale of the sampling distribution for the random-feature approximation ($p(\omega)$), where larger values emphasize broader alignment and smaller values increase sensitivity to finer geometric structure. For this reason, when the initial alignment is poor, the paper proposes applying MMD-Reg sequentially with decreasing values of $\ell$, using the previous estimate to initialize the next stage. The sequential schedule used on Wild Places is therefore intended as a coarse-to-fine refinement strategy, analogous in spirit to multi-scale ICP, and we chose the decreasing $\ell$ values to mirror the progressively finer alignment regime used by the ICP baseline in that experiment. We will revise the manuscript to make this distinction clearer and to state more explicitly that the current work does not claim global registration guarantees for the non-convex objective.
>
> We also note that the paper already includes a learned initialization mechanism for local MMD-Reg. In Neural MMD-Reg, a set transformer predicts an initial transformation, which is then refined by the differentiable MMD-Reg layer. In the supervised setting, it also predicts the kernel scale $\ell$ and, for partial-overlap settings, pointwise overlap weights. This was intended to show that MMD-Reg can serve both as a standalone local solver and as a learned refinement module when stronger initialization is needed.
>
> Regarding the scalability claim, we appreciate the opportunity to clarify the intended scope. The main claim is algorithmic: for fixed $D$, the random-feature surrogate reduces the quadratic kernel computation to linear complexity in the number of points by working with feature means rather than pairwise kernel sums. On PCPNet, Figure 2 examines runtime and accuracy as the number of points increases, showing gradual CPU scaling and near-constant GPU runtime for MMD-Reg. We will revise the discussion to make this distinction between the algorithmic claim and the empirical scaling study more explicit.
>
> We thank the reviewer again for the constructive comments and the positive overall recommendation. The points raised are very helpful in clarifying where the empirical presentation can be strengthened without changing the core claims of the paper. We believe these revisions will make the paper's contribution and intended scope more clear.

---

> > ### Author Rebuttal · Reviewer_rdAv · 2026-04-03
> >
> > Thanks for the authors' response and addressed my concern, so I keep my positive scores.

---

### Official Review · Reviewer_F61L · 2026-03-15

**Soundness:** 3
**Presentation:** 2
**Significance:** 2
**Originality:** 3
**Overall Recommendation:** 3
**Confidence:** 4

**Summary:**

This paper introduces MMD-Reg, a correspondence-free and differentiable approach to rigid 3D point-cloud registration. The method frames alignment as a nonlinear least-squares problem that minimizes the Maximum Mean Discrepancy (MMD) between point distributions, utilizing random Fourier features (RFF) to approximate the kernel. The objective is optimized using the Levenberg-Marquardt algorithm, and the resulting transformation is made differentiable via the implicit function theorem (IFT). This formulation allows MMD-Reg to operate as a plug-and-play optimization layer within deep learning architectures, which the authors demonstrate by integrating it with a Set Transformer for both supervised and unsupervised training paradigms

**Compliance With Llm Reviewing Policy:**

Affirmed.

**Key Questions For Authors:**

None

**Limitations:**

yes

**Strengths And Weaknesses:**

Strengths: The correspondence-free, probabilistic nature of the MMD objective makes the standalone registration inherently more robust to outliers, heavy noise, and non-uniform sampling densities compared to standard geometric baselines like ICP and GICP.

Weaknesses
1. Inadequate Baselines in Real-World Scenarios: In the evaluation on the large-scale, unstructured Wild Places dataset, MMD-Reg is compared exclusively against classic Multi-Scale ICP variants (Pt2Pt and Pt2Pl). Omitting modern learning-based registration networks or advanced probabilistic baselines in this crucial real-world experiment prevents a rigorous assessment of where the method stands against current state-of-the-art systems.

2. High Non-Convexity and Initialization Dependence: The objective function is highly non-convex due to its reliance on the rotation matrix and the trigonometric functions inherent in the RFF mapping. As a result, the Levenberg-Marquardt solver is prone to local minima and relies heavily on heuristics like sequential scale parameter reduction or a dedicated neural network initializer. The framework masks the underlying optimization landscape issues rather than resolving them mathematically.

3. Incremental Theoretical Innovation: The core technical pipeline is primarily a systematic integration of well-established components: MMD with RFF for linear scaling , the JAXopt library for implicit differentiation , and existing Set Transformer architectures for feature extraction. While functional from an engineering standpoint, the algorithmic novelty regarding 3D spatial transformations or point cloud probability measures is marginal for an ICML submission.

---

> ### Author Rebuttal · Authors · 2026-03-31
>
> We thank the reviewer for the careful reading of the paper and the clear summary of the method. We also appreciate the positive assessment of the robustness benefits of the correspondence-free formulation. Our goal in this work is to present a registration objective that is correspondence-free, scalable via random-feature approximation, and differentiable, enabling its use as an optimization layer within learning-based pipelines.
>
> Regarding originality, we agree that the individual components (MMD, random Fourier feature approximation, implicit differentiation, and set transformers) are well established. Our contribution lies in their formulation and integration in the context of rigid point-cloud registration: expressing registration as minimization of a random-feature surrogate of MMD, observing that this yields a standard nonlinear least-squares problem over rigid transformations, and enabling its use as a differentiable optimization layer via implicit differentiation. We will revise the introduction and related-work discussion to better reflect this scope and positioning.
>
> More broadly, we agree that the paper is strongest as an algorithmic and empirical contribution rather than as a contribution focused on new optimization theory. The primary evidence we provide is that the formulation yields accurate and scalable standalone registration under noise, density imbalance, and large-scale real LiDAR scenes in unstructured natural environments, while also supporting differentiable refinement in supervised and unsupervised learning settings. We will revise the manuscript to clarify the intended contribution and empirical scope, including a revised discussion of real-world baselines and of the role of initialization in the non-convex objective.
>
> On the question of non-convexity and initialization dependence, we agree that the objective is non-convex. The paper does not claim otherwise, nor does it claim global convergence guarantees for the Levenberg--Marquardt solver. More generally, non-convexity is an inherent challenge in rigid registration. Constraints on rotations in $SO(3)$, discrete correspondence variables in ICP-type methods, and non-convex objectives in learning-based approaches all contribute to non-convex optimization landscapes. As is standard in this literature, there is a trade-off between computationally expensive global search strategies and more efficient local optimization methods that rely on initialization and heuristics.
>
> Our contribution is not to resolve this trade-off theoretically, but to present a correspondence-free nonlinear least-squares objective together with practical mechanisms that make it effective despite non-convexity. In the standalone setting, we propose sequential application with decreasing kernel scales $\ell$. In the learning-based setting, Neural MMD-Reg predicts an initialization and, in the supervised partial-overlap regime, additionally predicts the kernel scale and pointwise overlap weights used inside the MMD objective.
>
> We appreciate the concern that these mechanisms could be interpreted as masking optimization difficulty. Our intention instead is to demonstrate that, despite the non-convex landscape, the formulation is practically useful in two complementary regimes: as a standalone local solver on large point clouds, and as a differentiable refinement layer in end-to-end trainable models. This distinction is reflected explicitly in the experiments: PCPNet and Wild Places evaluate standalone MMD-Reg, while ModelNet40 evaluates the learned pipeline and additionally reports set-transformer-only baselines to isolate the effect of the MMD-Reg layer relative to the learned initializer. We will make this separation clearer in the revision.
>
> Regarding the Wild Places evaluation, we agree that broader real-world comparisons could further strengthen the paper. The intent of that experiment was to evaluate standalone MMD-Reg on large-scale LiDAR scenes in unstructured natural environments, and for that reason we compared against strong geometric baselines operating in the same local-registration regime, namely multi-scale ICP point-to-point and point-to-plane variants. Learning-based registration methods are evaluated separately on ModelNet40, where Neural MMD-Reg couples MMD-Reg with a set transformer and is evaluated in both unsupervised full-overlap and supervised partial-overlap settings. We will clarify this experimental design choice in the revision.
>
> We thank the reviewer again for the constructive feedback. The points raised are helpful in clarifying where the presentation and evaluation can be strengthened without changing the core contributions of the paper. We believe these clarifications will make the paper's scope, claims, and intended use cases clearer to the reader.

---

> > ### Author Rebuttal · Reviewer_F61L · 2026-04-07
> >
> > I have carefully read the authors' rebuttal. While I acknowledge the effort to clarify the paper's scope as an empirical and engineering contribution, the rebuttal does not resolve my fundamental concerns regarding the paper's suitability for publication.Incremental Contribution: The authors concede that the core components (MMD, RFF, IFT, Set Transformers) are well-established and frame their contribution as merely the integration of these tools. For a premier machine learning conference, simply assembling existing techniques without resolving the underlying theoretical bottlenecks (e.g., the non-convex optimization landscape) or introducing a fundamentally novel mechanism is too incremental.Over-reliance on Initialization: The rebuttal acknowledges the inherent non-convexity of the MMD objective and relies heavily on initialization strategies—either a sequential heuristic of decreasing kernel scales or a pre-alignment from a Set Transformer. This essentially shifts the burden of the core registration challenge to the initializer. In the learning-based pipeline, if the Set Transformer provides a sufficiently accurate initial alignment, the actual performance gain attributable to the MMD-Reg refinement layer is marginal. The experiments on the ModelNet40 dataset  are insufficient to prove that the MMD-Reg objective can independently drive convergence from poor initializations in complex real-world geometries.Insufficient Baselines on Real-World Data: The authors defend their choice to exclusively compare against multi-scale ICP variants on the Wild Places dataset  by artificially restricting the scope to a "local-registration regime." A newly proposed registration objective that claims superior robustness and scalability must be benchmarked against modern, state-of-the-art robust registration methods (both correspondence-free and learning-based), not just classical ICP.

---

> > > ### Author Response · Authors · 2026-04-08
> > >
> > > We thank the reviewer for the time spent reading the paper and for articulating the remaining concerns. We appreciate the opportunity to further clarify the intended contribution and the evidence provided in the manuscript.
> > >
> > > We believe the paper is more than an empirical or engineering contribution because its central contribution is methodological: it introduces a distinct registration formulation in which rigid alignment is posed as minimization of a random-feature surrogate of MMD over rigid transformations, yielding a correspondence-free nonlinear least-squares objective that is scalable and differentiable. The novelty of the paper therefore lies in the registration formulation itself. This is more than a simple combination of existing components, because the formulation is specifically designed to avoid hard nearest-neighbor correspondences and explicit parametric density fitting, while remaining scalable through random features and differentiable through implicit differentiation. These properties are central to the paper because they allow the same objective to be used both as a standalone registration solver and as a differentiable refinement layer in learning-based pipelines. More precisely, the contribution is the combination of correspondence-free alignment, linear scaling for fixed feature dimension, and differentiable optimization that is not offered by the compared alternatives. We believe this combination is of direct interest to ML because it connects distribution matching, differentiable optimization, and geometric learning in a single formulation. This is also reflected in the experimental structure of the paper: PCPNet and Wild Places evaluate the standalone objective without a learned initializer, while ModelNet40 evaluates the same formulation as a differentiable refinement layer. In the latter case, set-transformer-only baselines are included to isolate the contribution of the MMD-Reg layer, which is substantial in the unsupervised clean setting and measurable in the supervised partial-overlap setting.
> > >
> > > Non-convexity is inherent to rigid registration, including correspondence-based methods such as ICP and learning-based registration pipelines. The practical question we consider is therefore, under matched initialization and evaluation conditions, whether the objective yields accurate solutions in the regimes studied. This is the setting considered in our standalone local-registration experiments on PCPNet and Wild Places, where the compared methods all begin from the identity transformation. Under these matched conditions, MMD-Reg attains lower registration error than the compared baselines. Our point is therefore not that the proposed formulation eliminates non-convexity, but that it yields a correspondence-free objective that is effective in practice and produces better solutions in the evaluated settings than the compared alternatives from the same initialization. We further emphasize that these standalone results do not rely on a learned initializer.
> > >
> > > We also note that the role of initialization is further clarified by the structure of the experiments in the paper. In the standalone PCPNet and Wild Places experiments, MMD-Reg is not preceded by a learned initializer. On Wild Places, the sequential reduction in kernel scale is part of the proposed standalone formulation, since the scale parameter directly controls whether the objective emphasizes coarse or fine geometric structure. In the learned setting, we explicitly report set-transformer-only baselines to isolate the effect of the MMD-Reg layer. These design choices are intended precisely to separate the roles of standalone optimization and learned initialization, rather than to attribute all registration performance to the initializer. We also note that, as mentioned in the paper, the LM solver was empirically reliable for this formulation: the same initial damping parameter was used across all settings and was found to work well throughout. This is intended only to clarify that the reported performance does not rely on extensive per-dataset solver tuning.
> > >
> > > Regarding the reviewer's concern about real-world convergence from poor initialization, we agree that the ModelNet40 experiments are not intended to establish that claim on their own. In the paper, that question is addressed more directly by the standalone Wild Places experiments, where MMD-Reg is not preceded by a learned initializer and begins from the identity transformation on large-scale outdoor LiDAR scans in unstructured natural environments. We therefore view Wild Places as evidence that the proposed objective can drive convergence in challenging real-world geometries within the local-registration regime studied in the paper. ModelNet40 serves a different purpose: it is the benchmark on which we compare against a broad set of recent state-of-the-art learning-based registration methods and evaluate the MMD-Reg layer as a differentiable refinement module.

---

### Decision · Program_Chairs · 2026-04-30

**Decision:**

Accept (regular)

**Comment:**

The paper presents a clear and technically solid method for correspondence free point cloud registration that is scalable, differentiable, and empirically strong across synthetic and real data. The main strengths are the elegant formulation, good presentation, and convincing evidence that the method can work both as a standalone solver and as a differentiable layer. The main reservations are that the core novelty is viewed by some reviewers as more integrative than fundamental, and that the empirical study would be stronger with broader real world baselines and deeper ablations on random feature choices and robustness to harder initialization. Overall, the rebuttal addressed most reviewer concerns and the balance of opinions is positive.